# Differences between Rural and Urban Practices in the Response to the COVID-19 Pandemic: Outcomes from the PRICOV-19 Study in 38 Countries

**DOI:** 10.3390/ijerph20043674

**Published:** 2023-02-19

**Authors:** Ferdinando Petrazzuoli, Claire Collins, Esther Van Poel, Athina Tatsioni, Sven Streit, Gazmend Bojaj, Radost Asenova, Kathryn Hoffmann, Jonila Gabrani, Zalika Klemenc-Ketis, Andrée Rochfort, Limor Adler, Adam Windak, Katarzyna Nessler, Sara Willems

**Affiliations:** 1Department of Clinical Sciences, Centre for Primary Health Care Research, Lund University, 21428 Malmö, Sweden; 2Research Centre, Irish College of General Practitioners, D02 XR68 Dublin, Ireland; 3Department of Public Health and Primary Care, Ghent University, 9000 Ghent, Belgium; 4Research Unit for General Medicine and Primary Health Care, Faculty of Medicine, School of Health Sciences, University of Ioannina, 45110 Ioannina, Greece; 5Institute of Primary Health Care (BIHAM), University of Bern, Mittelstrasse 43, 3012 Bern, Switzerland; 6Department of Management of Health Services and Institution, Heimerer College, 1000 Pristina, Kosovo; 7Department of Urology and General Practice, Faculty of Medicine, Medical University Plovdiv, 4003 Plovdiv, Bulgaria; 8Department of Social- and Preventive Medicine, Medical University of Vienna, 1090 Vienna, Austria; 9Faculty of Medicine, University of Basel, 4001 Basel, Switzerland; 10Department of Family Medicine, Medical Faculty, University of Maribor, Tabroska 8, 2000 Maribor, Slovenia; 11Department of Family Medicine, Medical Faculty, University of Ljubljana, Poljanski Nasip 58, 1000 Ljubljana, Slovenia; 12Ljubljana Community Health Centre, Metelkova 9, 1000 Ljubljana, Slovenia; 13Department of Family Medicine, Sackler Faculty of Medicine, Tel Aviv University, Tel Aviv 6195001, Israel; 14Department of Family Medicine, Jagiellonian University Medical College, 31-061 Krakow, Poland

**Keywords:** primary health care, general practice, urban, rural, quality of care, international comparison, COVID-19, PRICOV-19

## Abstract

This paper explores the differences between rural and urban practices in the response to the COVID-19 pandemic, emphasizing aspects such as management of patient flow, infection prevention and control, information processing, communication and collaboration. Using a cross-sectional design, data were collected through the online PRICOV-19 questionnaire sent to general practices in 38 countries. Rural practices in our sample were smaller than urban-based practices. They reported an above-average number of old and multimorbid patients and a below-average number of patients with a migrant background or financial problems. Rural practices were less likely to provide leaflets and information, but were more likely to have ceased using the waiting room or to have made structural changes to their waiting room and to have changed their prescribing practices in terms of patients attending the practices. They were less likely to perform video consultations or use electronic prescription methods. Our findings show the existence of certain issues that could impact patient safety in rural areas more than in urban areas due to the underlying differences in population profile and supports. These could be used to plan the organization of care for similar future pandemic situations.

## 1. Introduction

The COVID-19 pandemic is the largest disastrous health crisis ever experienced in recent centuries, as it has confronted health systems with extraordinary challenges, often placing extreme pressure on the health workforce and requiring rapid changes in their deployment, with rural and deprived areas worst-served [1].

During the lockdown in Italy, which was the first country in Europe to be seriously affected, health services were under severe strain, especially regarding their ability to provide adequate care to both COVID-19 patients and other patients. Outpatient secondary care services were closed to the public across the country and scheduled patient visits for non-life-threatening conditions were suspended. In this context, primary care doctors were called upon to manage an increasing number of healthcare situations by reorganizing their services and modifying their methods of providing care. Many primary care physicians quickly moved to remote consultations, although evidence-based local, regional and national guidelines on managing COVID-19 were lacking at the time. Services and the reorganization of care delivery were left to the capabilities of individual general practitioners [2].

Low- and middle-income countries have faced many challenges in controlling the COVID-19 pandemic; in these countries healthcare resources are limited, SARS COV 2 testing is conducted on a limited scale and treatment options are few. Very often there is no vaccine. Only low-cost solutions prevail for the prevention, diagnosis and treatment of SARS-CoV-2 [3].

General practitioners (GPs) played a crucial role in the fight against the COVID-19 pandemic. However, they have experienced many barriers to fulfilling this role and many domains of quality of care have been affected [4,5,6]. For example, a scoping review revealed that healthcare access and utilization for both COVID and non-COVID-19 services decreased almost universally, across both higher- and lower-income countries [7,8]. Cancellations of appointments, examinations and surgeries have been a constant negative effect of the pandemic [9].

Rural populations already experience worse health status than urban populations. This is partly due to a higher incidence of chronic conditions, higher age and vulnerability, higher child and maternal health problems and higher engagement in health risk behaviors [10]. At the same time, access to care is lower in rural settings due to factors such as travel distance, lack of internet broadband, provider shortages [11,12] and maldistribution of resources [12,13]. A rising concern in many European countries is the growing shortage of GPs, particularly in rural and remote regions. Whereas the overall number of doctors per capita has increased in nearly all countries, the per capita quota of GPs has decreased in most countries. On average across EU countries, only about one in five doctors were GPs in 2018 [7,14,15].

Additionally, populations in rural and remote areas may be more vulnerable to the consequences of extreme events due to geographical isolation, less-developed infrastructure, less epidemiological surveillance capacity and less favorable social determinants of health [16,17,18,19,20,21,22,23]. We know that in many countries GPs in rural areas already have a heavier workload than their counterparts in semi-urban and urban areas [23,24]. A study carried out in the USA showed widespread adverse secondary impacts from the COVID-19 pandemic, including to mental health, social relationships and financial well-being, but these consequences of the pandemic have not been distributed equally across geography [25].

Regarding prevention measures, a previous study showed that rural residents are significantly less likely to have worn a mask in public, sanitized their home or workplace with disinfectant, avoided dining at restaurants or bars or worked from home [26].

In some fields, rural practices seemed to have performed better than their urban counterparts. In a study carried out in New Zealand, a moderate degree of strain was experienced by general practices, although rural practices appeared to experience less strain compared with urban ones. Rural practices had fewer staff absent from work, were less likely to use alternative forms of consultations such as video consultations and telephone consultations and had possibly lower reductions in patient volumes. These variations might be related to personal characteristics of rural people as compared with urban practices or different models of care [27].

Given the above, it is reasonable to say that primary care professionals (PCPs) working in rural areas may face additional challenges in crises such as the COVID-19 pandemic. The question, therefore, arose regarding how well rural practices managed to organize care during the COVID-19 pandemic.

### Study Aim

This paper aims to describe the differences between rural and urban practices in the response to the COVID-19 pandemic. This paper hereby focuses on aspects of care such as patient flow management, infection prevention and control, information processing, communication and collaboration.

## 2. Methods

### 2.1. Study Design and Setting

In the summer of 2020, an international consortium of 45 research institutes was formed under the coordination of Ghent University (Belgium) to set up the PRICOV-19 study aiming to investigate how primary care (PC) practices were organized during COVID-19 to guarantee safe, effective, patient-centered and equitable care. The PRICOV-19 study also aimed to describe the association between the response to the pandemic and practice and healthcare system characteristics [28]. The questionnaire was developed in multiple phases, including a pilot study in Belgium. The final version includes 53 items divided into six sections: patient flow (including appointments, triage and management for routine care); infection prevention; information processing; communication; collaboration and self-care; and practice and participant characteristics. Using a cross-sectional design, data were collected in 37 European countries and Israel. More information on the study protocol is described elsewhere [28].

### 2.2. Measurements

Data were collected by means of an online self-reported questionnaire among PC practices. The questionnaire was developed at Ghent University in multiple phases, including piloting, and was translated into 38 languages following a standard procedure. The Research Electronic Data Capture (REDCap) platform at Ghent University was used to host the questionnaire in all languages, send out invitations to the national samples of PC practices and securely store the answers from the participants [29].

### 2.3. Sampling and Recruitment

Data were collected between November 2020 and December 2021, except for in Belgium, where data were partially collected earlier. Data collection varied in duration between countries from three to 35 weeks. In each partner country, the consortium partner(s) recruited PC practices following a pre-defined recruitment procedure. Drawing a randomized sample among all PC practices in the country was preferred over convenience sampling. Partners logged all the steps taken in the sampling procedure. PRICOV-19 aimed to sample between 80 and 200 PC practices per country, depending on the national number of PC practices. However, since there was no funding for this study and coordinators recruited practices voluntarily, it was impossible to enforce a specific recruitment strategy or specific response rates. Per practice, one questionnaire was completed, preferably by a GP or by a staff member familiar with the practice organization. The overall response rate was 22.0%.

### 2.4. Variables

#### Practice Location

In all countries except for Belgium, the practice location was determined based on the responses to the following survey item: How would you characterize the place of this practice? The original five answer options (big (inner)city, suburbs, (small) town, mixed urban–rural, rural) were recoded into a dichotomous variable to represent the rural PC practices. PC practices located in a big (inner)city, suburbs or (small) town were considered as urban, while the other categories were combined to rural. For Belgium, the rurality of the practice location was based on the ZIP code considering the population density.

### 2.5. Description of the Sample

The sample was described based on the number of participants per country (see Figure 1) and five practice characteristics: the number of GPs (median value, irrespectively of the full-time equivalent); number of paid staff members (median value, irrespectively of the full-time equivalent); being a teaching practice for GP trainees (yes); capitation payment system (yes); and the patient population composition (having an above-average vulnerable patient population). For the latter, a distinction was made between patients with chronic conditions, patients over the age of 70, patients with low (health) literacy, patients with a migration background with difficulty speaking the local language, financial problems, a psychiatric vulnerability and little social support or limited informal care.

### 2.6. Outcome Variables

Twenty-four survey questions were selected as outcome variables in the following themes: collaboration with other practices and experienced support (#4), the involvement of non-GP staff (#2), patient safety incidents during COVID-19 (#5), the use of protocols (#2), information for patients (#4) and initiatives for vulnerable patients (#7). A description of these survey questions and their original answer options is provided in Appendix A.

### 2.7. Data Analysis

Ghent University was responsible for cleaning all data using SPSS Statistics for Windows, version 28.0 (IBM Corp., Armonk, NY, USA). Statistical analysis was performed using SPSS software (version 28.0 SPSS Inc., Illinois) on Version 8 of the database (cleaned data of 38 countries available as of 14 February 2022). Descriptive analysis was undertaken to describe the different characteristics and measures taken during COVID-19 between urban and rural practices. Continuous variables are presented as mean and standard deviation (SD) if normally distributed and median and interquartile range (IQR) if not. Categorical variables are presented by numbers (n) and valid percentages (%). Answer options of “I do not know” and “not applicable” were considered as invalid. When investigating the differences between urban and rural practices, a Fisher’s exact test was computed for binary variables, a Mann–Whitney U test was conducted to compare non-normally distributed numerical variables and an independent two-sided t-test for numerical continuous variables was performed, with *p* < 0.05 considered statistically significant.

### 2.8. Ethical Approval

The Research Ethics Committee of Ghent University Hospital approved the protocol of the PRICOV-19 study (BC-07617). Research ethics committees in the different partner countries gave additional approval if needed in that country. All participants gave informed consent on the first page of the online questionnaire.

## 3. Results

The analysis included 5539 practices from 38 countries. Figure 1 shows the geographic variation in the proportion of practices which were rural, ranging from 11.1% in Romania to 64.9% in Greece and 37.5% among the overall responding sample.

On average, practices counted three GPs and seven paid staff members. Just under half of the practices were teaching practices for GP trainees (47.5%). The majority of the practices had a capitation payment system (57.0%).

Table 1 describes the differences between the participating urban and rural practices in terms of practice characteristics and patient population. It shows that rural practices in our sample were significantly smaller in terms of number of GPs, were significantly less often a training practice and were more often a capitation type payment system. Rural practices were significantly more likely to report an above-average number of patients with chronic conditions and patients aged over 70 years. On the other hand, they were significantly less likely to report an above-country average number of patients with a migrant background and patients with financial problems. No significant difference between urban and rural practices was noted in terms of other patient groups.

Table 2 describes how practices collaborated and the support they experienced during the COVID-19 pandemic. No differences were observed between urban and rural practices in terms of experienced support from other practices or the ability to redistribute tasks if the staff was absent because of COVID-19.

Rural practices were more likely to have GP support always available if needed by staff doing telephone triage.

In terms of the possible occurrence of patient safety incidents, rural practices were significantly more likely to report that at least one patient safety incident occurred due to patient factors, while a similar difference was not observed in terms of practice-related factors.

Concerning measures to safeguard staff, rural practices were significantly more likely to have stopped using the waiting room, made structural changes to their waiting room and changed their prescribing practices in terms of patients attending the practices. However, there was no difference in the proportion of urban and rural practices that reported big limitations to the provision of high-quality care due to their building/infrastructure. 

Rural practices were significantly less likely to perform video consultations or use electronic prescription methods. 

Rural practices were significantly less likely to have a protocol for answering calls from potential COVID-patients. While the overall proportions allowing enough time to disinfect rooms between consultations were low at under one-third of practices, no difference between urban and rural practices was observed.

Patient information and outreach activities are shown in Table 3. Rural practices were significantly less likely to provide leaflets and information on their website and answering machine in multiple languages. 

In terms of patient outreach activities, no significant differences were noted between urban and rural practices. Other than for patients with chronic conditions, outreach activities were low for all groups listed.

The most relevant findings on the differences between urban and rural settings are summarized in Table 4.

## 4. Discussion

### 4.1. Summary of the Findings

The results of PRICOV-19 showed that the COVID-19 pandemic significantly impacted the organization of primary care practices. Practice location is shown to be an actor in how general practices across 38 countries responded to the COVID-19 pandemic, and this applies for all domains of care organization: patient flow (including appointments, triage and management for routine care); infection prevention; information processing; communication; collaboration and self-care; and practice and participant characteristics. Rural practices were more likely to have GP support always available if needed by staff when doing telephone triage. When measures to safeguard staff are considered, rural practices were significantly more likely to have ceased use of the waiting room, to have made structural changes to their waiting room and to have changed their prescribing practices in terms of patients attending the practices. Rural practices were significantly less likely to be performing video consultations or using electronic prescription methods. Rural practices were also significantly less likely to have a protocol for answering calls from potential COVID-19 patients and to provide leaflets and information on their website and answering machine in multiple languages.

The degree of changes adopted by GP practices across countries indicates a strong ability to adapt, as confirmed in other studies [30]. However, as our data also showed, there was great variability in the speed and scope of the responses to the impact of COVID-19 in the practices, indicating potentially variable standards and quality of care [31].

It has been reported that, although with limited resources and support, the rural doctors’ practical responses to the COVID-19 crisis underscore strong problem-focused coping strategies and shared commitments to their communities, patients and colleagues [28]. When measures to safeguard staff are considered, rural practices were more significantly likely to have stopped use of the waiting room, to have made structural changes to their waiting room and to have changed their prescribing practices in terms of patients attending the practices. However, rural practices were significantly less likely to have a protocol for answering calls from potential COVID-patients.

The literature shows that, globally, chronic disease monitoring was postponed, with possible consequences in the course of disease of patients [32,33,34,35]. According to Windak et al. [36], family physicians experienced that acute care was compromised because patients consulted practices less frequently for non-COVID-19 problems. Monitoring visits were postponed or canceled and screening examinations for the early detection of chronic diseases were particularly neglected [37,38,39].

Our results here show that outreach activities were low for all patient groups, including those with chronic conditions, regardless of the rurality level. In our study, during the pandemic, a list of chronic patients was compiled by less than one-third of responding practices. A potential explanation for this issue might be that PC practices in several countries or regions may not have electronic medical records (EMR). EMR can be used to pre-screen patient needs and identify high-risk patients and those with gaps in care while allowing multidisciplinary teams to coordinate care and co-manage patients with complex needs [39,40].

Access to care directly impacts one’s overall physical, social and mental health status and quality of life. Barriers to access to care often vary based on socioeconomic status, ethnicity, age, sex, disability status and residential location. These barriers are more prevalent among vulnerable communities [7,41,42,43,44]. The discrepancies in access to care among vulnerable communities and their attitudes toward and acceptance of pandemic mitigation measures, as well as triage policies used during the pandemic, are not well understood [45]. There were no significant differences between urban and rural practices on outreach to vulnerable patients. With hindsight, we have learned that intimate partner violence (IPV) is one specific area that can be actioned in future pandemics [46]. Regarding patients with previous problems of domestic violence or with a problematic child-rearing situation, less than one-quarter of practices adopted a proactive strategy to contact these patients, with no statistically significant differences between rural and urban practices. However, there is an argument to indicate that such strategies may be especially necessary in rural settings, with a recent review of the potential implications of COVID-19 on intimate partner violence (IPV) risk globally determining that social and geographical isolation increased risk for IPV [46]. This conclusion is consistent with Edwards’ seminal review of the urban/rural IPV divide [47], which found that rural women experienced more chronic and severe IPV and had more severe psychosocial and physical health outcomes, including increased post-traumatic stress disorder (PTSD), due to the lack of IPV-related services. In addition, the low percentage of practices adopting a proactive strategy may imply that GP practices lack sufficient motivation, or training, to implement this action in everyday practice [48]. This is one example of how proactive planning by policymakers and family medicine leaders can improve the quality of care for specific patient groups in the event of future pandemics. Unfortunately, in many countries, there is already a GP workforce shortage, and when staff resources were deployed to manage the pandemic workload, it was at the expense of chronic disease management, IPV and potential cancer presentations [49]. The need for further training of PC professionals in screening and providing collaborative care for vulnerable patients should be included among essential clinical skills.

In line with Due et al. 2021 [50] and Petrazzuoli et al. [51], the use of alternative consultation forms seems context-bound, influencing one’ willingness to use these alternatives. Rural practices were significantly less likely to perform video consultations or use electronic prescription methods. There is evidence that remote consultations can help in delivering high-quality care during a pandemic [52]. Therefore, it is important that in rural areas, the conditions are established so that this could be achieved. More than one-third of residents in rural areas of the US reported that broadband and computer access were significant obstacles to utilizing telehealth [53]. Apart from the digital divide, other factors are at play here, possibly health literacy and older populations [54]. These and other factors need to be addressed to improve the triage of patients in rural practices in pandemic situations.

The COVID-19 pandemic also created new problems with patient safety at multiple system levels. Before the pandemic, according to the OECD, globally, as many as four in ten patients were harmed in primary and outpatient health care [15]. Up to 80% of harm was preventable [55]. It is estimated that 8–12% of patients admitted to a hospital in the EU suffer from adverse effects [56]. There is concern about the ethics of triage, allocation of scarce resources and decision-making during pandemics [57]. In our study, while practices reported that safety incidents did occur, we did not see significant differences between urban and rural practices, except in relation to those related to patient factors. Reports of such safety incidents were higher in rural practices, where there were also more likely to be a higher-than-average number of patients over 70 years. 

Rural practices performed as well as urban practices on several parameters, and in some cases performed better than their urban counterparts. The strengths of rural practices are clearly demonstrated and worthy of further exploration.

### 4.2. Strengths and Limitations

The strengths of this study are its wide coverage of GP respondents across 38 countries and the wide coverage of different COVID-related issues in the questionnaire. Additionally, the questionnaire was developed and validated in several phases, including a pilot study in Flanders.

The limitations include the relatively low response rate (22.0%), with considerable differences between countries. Moreover, the survey was based on a self-selecting sample, which comes with inherent bias. Additionally, the sample was obtained differently depending on the participating country. Moreover, the data collection was carried out over several months, therefore covering different phases of the pandemic. In addition, rural practices may be differently characterized in different countries, which may influence the results of our study. Moreover, the practice location was indicated based on the subjective perception of the respondent, except for Belgium.

## 5. Conclusions

Differences in family medicine practices’ organization between rural and urban areas indicate the need to take into account these differences when preparing plans for future similar situations. Although the issues that rural family medicine practices were faced with during the COVID-19 pandemic are in some points similar to the issues in urban practices (such as postponing consultations and limited resources), there are still some issues that could impact the safety of the patients in rural areas more than those in urban areas. Our results underline the need to further explore the underlying factors for rural/urban differences and use the findings to plan the care organization for future similar pandemic situations. Particular emphasis should be placed on reducing inequalities in access to modern forms of remote communication with the use of computer and ICT techniques. Improvement in this area is needed in both urban and rural practices, but there is a greater need in the latter. Furthermore, it appears from our data that GP practices, regardless of urban/rural location, responded according to recommendations during the COVID-19 pandemic, but differences in patient populations may require differential responses which need to be factored in. Our findings show the existence of certain issues that could impact patient safety in rural areas more than in urban areas due to the underlying differences in population profile and supports. These could be used to plan the organization of care for similar future pandemic situations.

## Figures and Tables

**Figure 1 ijerph-20-03674-f001:**
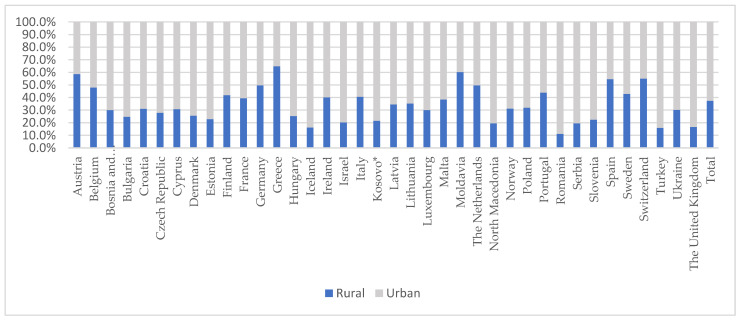
Proportion of rural and urban practices for each participating country. * All references to Kosovo, whether the territory, institutions, or population, in this project shall be understood in full compliance with the United Nations Security Council Resolution 1244 and the ICJ Opinion on the Kosovo declaration of independence, without prejudice to the status of Kosovo.

**Table 1 ijerph-20-03674-t001:** Comparison of practice and population profile between urban and rural practice (*n* = 5539).

Characteristic	Urban	Rural	*p* Value
**Number of GPs in practice (*n* = 5437)** Median (IQR)	3 (1–5)	2 (1–4)	<0.001 ^1^
**Number of paid staff members (*n* = 5434)** Median (IQR)	7 (3–19)	8 (3–18)	0.583 ^1^
**Being a teaching practice for GP trainees (*n* = 5144)** Yes	49.9%	43.7%	0.001 ^2^
**Capitation payment system (*n* = 5512)** Yes	47.6%	36.8%	0.001 ^2^
**Patients with chronic conditions (*n* = 5350)** Above-average number	39.0%	42.7%	0.008 ^2^
**Patients over the age of 70 (*n* = 5392)** Above-average number	37.3%	44.6%	0.001 ^2^
**Patients with limited health literacy (*n* = 5222)** Above-average number	18.9%	19.5%	0.637 ^2^
**Patients with a migration background (*n* = 5159)** Above-average number	23.1%	11.4%	0.001 ^2^
**Patients with financial problems (*n* = 5257)** Above-average number	25.1%	22.4%	0.026 ^2^
**Patients with a psychiatric vulnerability (*n* = 5242)** Above-average number	19.6%	18.4%	0.278 ^2^
**Patients with little social support (*n* = 5135)** Above-average number	20.3%	19.1%	0.296 ^2^

^1^ Mann–Whitney U test; ^2^ Fisher’s exact test; (IQR): interquartile range.

**Table 2 ijerph-20-03674-t002:** Practice organization (*n* = 5539).

	Urban	Rural	*p* Value
Collaboration with other practices and experienced support			
If staff members in this practice are absent because of COVID-19 (infection or quarantine), this practice can count on the help of other PC practices in the neighborhoods	42.4%	42.0%	0.805 ^2^
The COVID-19 pandemic has promoted cooperation with other PC practices in the neighborhoods	37.3%	37.8%	0.731 ^2^
The practice experienced large limitations related to the building/infrastructure in terms of providing high-quality and safe care	22.0%	21.5%	0.660 ^2^
If staff members in this practice are absent because of COVID-19 (infection or quarantine), the work can be distributed in such a way that the well-being of colleagues is not compromised	44.9%	43.3%	0.273 ^2^
The involvement of non-GP staff			
Staff members are more involved in giving information and recommendations to patients contacting the practice by phone	81.1%	82.9%	0.134 ^2^
In the situation where telephonic triage is performed by someone other than a GP in this practice and he/she needs support when assessing a call, he/she can always rely on support from a GP	67.7%	71.9%	0.003 ^2^
Patient safety incidents during COVID-19			
A patient with a fever caused by an infection other than COVID-19 was seen late because the COVID-19 protocol was followed which delayed the care	40.5%	38.8%	0.269 ^2^
A patient with an urgent condition was seen late because he/she did not come to the practice sooner	59.4%	62.2%	0.071 ^2^
A patient with a serious condition was seen late because he/she did not know how to call on a GP	26.1%	32.0%	<0.001 ^2^
A patient with an urgent condition was seen late because the situation was assessed as non-urgent during the telephonic triage	21.6%	19.1%	0.056 ^2^
A patient with an urgent condition other than COVID-19 was assessed incorrectly during the triage procedure	28.5%	27.6%	0.553 ^2^
The use of protocols			
A protocol is used in this practice when answering phone calls from potential COVID-19 patients	77.3%	74.2%	0.014 ^2^
Providing enough time to disinfect the room between consultations	29.9%	30.1%	0.924 ^2^

^1^ Mann–Whitney U test; ^2^ Fisher’s Exact test; PC = primary care; % = percentage of positive responses.

**Table 3 ijerph-20-03674-t003:** Information for patients and taking care of vulnerable patients (*n* = 5539).

	Urban	Rural	*p* Value
Information for patients			
Does this practice have a leaflet with information on COVID-19 to give to patients No Yes, in one language Yes, in multiple languages	49.8% 34.4% 15.8%	44.6% 40.7% 14.7%	<0.001 ^3^
Does the answering machine of this practice provide information in multiple languages?	11.3%	7.3%	<0.001 ^2^
Is the leaflet of this practice available to patients in multiple languages?	23.3%	19.2%	<0.001 ^2^
Is the information on the website of this practice available in multiple languages?	23.7%	12.9%	<0.001 ^2^
Vulnerable patients			
Since the COVID-19 pandemic, a list was compiled from the EMR for at least one group of patients with a chronic disorder	31.5%	28.0%	0.635 ^2^
This practice contacted patients with a chronic condition who needed follow-up care.	63.2%	61.9%	0.385 ^2^
This practice contacted psychologically vulnerable patients.	35.9%	35.4%	0.731 ^2^
This practice contacted patients with previous problems of domestic violence or with a problematic child-rearing situation	17.3%	17.0%	0.759 ^2^
When a patient needs to isolate him/herself, the extent to which this is feasible at his/her home is checked with the patient.	32.5%	33.6%	0.399 ^2^
Staff members are more involved in giving information or explaining what a caregiver has said to illiterate patients, patients with low health literacy or migrants	64.0%	64.1%	0.948 ^2^
Staff members are more involved in actively reaching out to patients that might postpone healthcare	57.0%	58.3%	0.368 ^2^

^1^ Mann–Whitney U test; ^2^ Fisher’s exact test; ^3^ chi-squared test; EMR = electronic medical record; % = percentage of positive responses.

**Table 4 ijerph-20-03674-t004:** Key findings of this paper at a glance.

Rural Practices More Likely than Urban	Rural Practices Similar to Urban	Rural Practices Less Likely than Urban
to have GP support always available if needed by staff doing telephone triage.	experience support from other practices	to perform video consultations or use electronic prescription methods
to have safeguarded staff through actions such as stopped using the waiting room, to have made structural changes to their waiting room and to have changed their prescribing practices in terms of patients attending the practices	able to redistribute tasks if staff are out sick	to have a protocol for answering calls from potential COVID-19 patients
to report that at least one patient safety incident occurred due to patient factors	to report that at least one patient safety incident occurred due to practice-related factors	to provide leaflets and information on their website and answering machine in multiple languages
	to reporting limitations to the provision of high-quality care due to their building/infrastructure	
	to provide enough time to disinfect rooms between consultations	
	to be able to adapt as practice	
	to organize patient outreach activities	

## Data Availability

The anonymized data are held at Ghent University and all raw data that could lead to the identification of the respondents were permanently removed. Reasonable request is required to access non-identifiable data by users who are external to the research teams involved. Access will be subject to a data transfer agreement and following approval from the principal investigator of the study.

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
