# Peer review of "Differences between Rural and Urban Practices in the Response to the COVID-19 Pandemic: Outcomes from the PRICOV-19 Study in 38 Countries"

_ijerph, 2023, doi:10.3390/ijerph20043674_

Round 1

Reviewer 1 Report

1-Review the title of work,  since objectively it is intended to evaluate the impacto of rurality on practices and COVID OR do you want to show  rural  practices con COVID wide  38 countries

2-In teh introduction  not  speak of rurality but of rural populations and health- vulnerabilty

3- Rurality is mesuared under territorial and environmental approach and the work  does include these parameters

4- The  work could hace greater impact if the  discussion were deepned  and intra e inter countries comparisons were made and generalities were  raised,  comparisons with other region outside european countries- other continents are  not made  despite  teh significance of the global  pandemic

5- Strategies or models for the improvement of these evaluated rural  practices are not  included, are not include management  approach, the discussion is very general general 

6-if the objetive  is rurality, these territorial or ambiental  variables  must  included in teh analysys.

7-another option is to direct the focus  od the article to quality  management of health services and covid, hospital management 

Author Response

Reviewer 1

Thank you very much for your suggestions which allow us to improve our manuscript.

Comments and Suggestions for Authors

1-Review the title of work, since objectively it is intended to evaluate the impact of rurality on practices and COVID OR do you want to show rural practices con COVID wide  38 countries

New Title

Differences between rural and urban practices  in the response  to the COVID-19 Pandemic: Outcomes from the PRICOV-19 Study in 38 Countries

2-In the introduction, do not speak of rurality but of rural populations and health- vulnerability

DONE changed where appropriate the word rurality.

3- Rurality is measured under territorial and environmental approach and the work does include these parameters

In the limitations: Moreover, the practice location was indicated based on the subjective perception of the respondent, except for Belgium - In our opinion this method is scientifically acceptable and used frequently in the literature .    Wilhelmi L, Ingendae F, Steinhaeuser J.  What leads to the subjective perception of a ‘rural area’? A qualitative study with undergraduate students and postgraduate trainees in Germany to tailor strategies against physician’s shortage. Rural and Remote Health 2018; 18: 4694. https://doi.org/10.22605/RRH4694.

4- The work could have greater impact if the  discussion were deepened  and intra e inter countries comparisons were made and generalities were  raised,  comparisons with other region outside European countries- other continents are  not made  despite  the significance of the global  pandemic.

As stated in the limitations: The limitations include the relatively low response rate (22.0%) with considerable differences between countries. Moreover, the survey was based on a self-selecting sample, which comes with inherent bias. Also, the sample was obtained differently depending on the participating country. Moreover, the data collection was carried out in several months therefore covering different phases of the pandemic. In addition, rural practices may be differently characterized in different countries, which may influence the results of our study. These factors do not allow for comparison between the countries. Anyway we have introduced in the introduction interesting outcomes from Non-European Countries.

5- Strategies or models for the improvement of these evaluated rural practices are not  included, are not include management  approach, the discussion is very general.

A full description of the Strategies or models for the improvement was out of the scope of the survey. As described in the article on the  protocol of the study,*  The PRICOV-19 study  is a big study which is primarily aimed at investigating how GP practices in 38 countries were organized during the COVID-19 pandemic to guarantee safe, effective, patient-centered, and equitable care. Also, the shift in roles and tasks and the wellbeing of staff members was researched. It was expected that both characteristics of the GP practice and health care system features were associated with how GP practices can cope with these challenges.The collection of data was not primarily aimed at exploring the impact of rurality on the practice management.

Van Poel, E.; Vanden Bussche, P.; Klemenc-Ketis, Z.; Willems, S., How did general practices organize care during the COVID-19 pandemic: the protocol of the cross-sectional PRICOV-19 study in 38 countries. BMC Primary Care 2022, 23, (1), 11.

6-if the objective  is rurality, these territorial or Ambiental  variables  must be include in the analysis.

As stated in point 4 the PRICOV-19 survey was primarily aimed at investigating how GP practices in 38 countries were organized during the COVID-19 pandemic to guarantee safe, effective, patient-centered, and equitable care. The practice location was based on the subjective perception of the respondent, except for Belgium (see also the limitations paragraph).

7-another option is to direct the focus of the article to quality management of health services and covid, hospital management.

Thank you for this suggestion. However, the survey was not aimed at exploring “quality management of health services and covid, hospital management.”

Reviewer 2 Report

This is an important and interesting study. The impact of rural location on GP or other health practice is of course crucial either for dealing with general health problems or the COVID-19 pandemic. The overall conclusions of the current study are plausible. However, the current study seems to have wasted this great chance to only report some minor findings. The analysis method can also be strengthened. Concretely:

1. The variables/factors leading to the difference between urban and urbal settings should be better discussed in the Introduction section. So far, only limited number of studies were cited to support the examination. I think this section can be expanded to not only clarify reasoning to investigate these factors but also consider more factors (and let us know why some important factors weren’t investigated)

2.  Overall, it seems that study hasn’t treated the pandemic as a great disastrous health crisis. It simply listed some factors having been cited in previous studies. However, COVID-19 should be different. For example, the quicker and greater number of deaths. These factors mean COVID-19 should be treated differently. Then, we want to know for these different treatment strategies (from regular health problems), what are the difference between rural and urban settings?

3. The study mainly covers European nations and Israel. What might be its implications for poorer nations?

4. The percentage of rural location of investigated samples is so different across countries, from Romania to Greece. But what this difference may mean to the investigated health services?

5. The authors mentioned intimate partner violence (IPV) and domestic violence, but didn’t provide any detailed discussion on the violence and the COVID-19 pandemic situation. Other psychological and mental issues weren’t widely covered either. Should the study cover/report more such information?

6. For analysis, would authors consider adopting more methods to reflect the possible interaction between the reported factors?

7. The study reported “Higher levels of rural broadband exist in Eu-rope, and in some countries, rural broadband coverage exceeds urban, indicating that 322 other factors are at play here, possibly health literacy and older populations” (P11, 321-323). Why didn’t the study provided more such information? Even though they were not covered by the survey, the authors can manage to get them from second-hand sources.

Finally, I don’t have public health practice experience and my academic training is health communication. So, if some of my suggestions, which are regular in health communication setting, may not be acceptable to public health scholarship, I am fine that the authors insist on their own preferences after providing appropriate explanations in responses or in manuscript revision. 

Author Response

Reviewer 2

Thank you very much for your suggestions which allow us to improve our manuscript

Comments and Suggestions for Authors

This is an important and interesting study. The impact of rural location on GP or other health practice is of course crucial either for dealing with general health problems or the COVID-19 pandemic. The overall conclusions of the current study are plausible. However, the current study seems to have wasted this great chance to only report some minor findings. The analysis method can also be strengthened. Concretely:

  1. The variables/factors leading to the difference between urban and urban settings should be better discussed in the Introduction section. So far, only a limited number of studies were cited to support the examination. I think this section can be expanded to not only clarify reasoning to investigate these factors but also consider more factors (and let us know why some important factors weren’t investigated).

Added in the INTRODUCTION AT LINE 84

  1. A study carried out in the USA showed widespread adverse secondary impacts from the COVID-19 pandemic, including to mental health, social relationships, and financial well-being but these consequences of the pandemic have not been distributed equally across geography. (REF: Monnat, Shannon M. “Rural-Urban Variation in COVID-19 Experiences and Impacts among U.S. Working-Age Adults.” The Annals of the American Academy of Political and Social Science vol. 698,1 (2021): 111-136. doi:10.1177/00027162211069717)

  1. Regarding prevention measures, a previous study showed that rural residents are significantly less likely to have worn a mask in public, sanitized their home or workplace with disinfectant, avoided dining at restaurants or bars, or worked from home. REF Callaghan, Timothy et al. “Rural and Urban Differences in COVID-19 Prevention Behaviors.” The Journal of rural health : official journal of the American Rural Health Association and the National Rural Health Care Association vol. 37,2 (2021): 287-295. doi:10.1111/jrh.12556
  2. In some fields rural practice seemed to have performed better than their urban counterparts. In a study carried out in New Zealand A moderate degree of strain was experienced by general practices, although rural practices appeared to experience less strain compared to urban ones. Rural practices had fewer staff absent from work, were less likely to use alternative forms of consultations such as video consultations and telephone consultations, and had possibly lower reductions in patient volumes. These variations might be related to personal characteristics of rural people as compared to urban practices or different models of care. [REF Eggleton, Kyle, Nam Bui, and Felicity Goodyear-Smith. "COVID-19 impact on New Zealand general practice: rural–urban differences." Rural and Remote Health 22.1 (2022).]

ADDED IN THE METHOD SESSION

The questionnaire was developed in multiple phases, including a pilot study in Belgium. The final version includes 53 items divided into six sections: patient flow (including appointments, triage, and management for routine care); infection prevention; information processing; communication; collaboration and self-care; and practice and participant characteristics.

  1. Overall, it seems that study hasn’t treated the pandemic as a great disastrous health crisis. It simply listed some factors having been cited in previous studies. However, COVID-19 should be different. For example, the quicker and greater number of deaths. These factors mean COVID-19 should be treated differently. Then, we want to know for these different treatment strategies (from regular health problems), what are the differences between rural and urban settings?

Added in the INTRODUCTION

The Covid 19 pandemic is the largest disastrous health crisis ever experienced in recent centuries.

Added in the INTRODUCTION

 During the lockdown, in Italy which was the first country in Europe to be seriously affected, health services have been under severe strain, especially regarding their ability to provide adequate care to both COVID-19 patients and other patients. Outpatient secondary care services were closed to the public across the country and scheduled patient visits for non-life-threatening conditions were suspended. In this context, primary care doctors  have been called upon to manage an increasing number of healthcare situations by reorganizing their services and modifying the methods of providing care. Many primary care physicians quickly moved to remote consultations, although evidence-based local, regional and national guidelines on managing COVID-19 were lacking at the time. Services and the reorganization of care delivery were left to the capabilities of individual general practitioners.

REF Kurotschka PK, Serafini A, Demontis M, et al. General Practitioners' Experiences During the First Phase of the COVID-19 Pandemic in Italy: A Critical Incident Technique Study. Front Public Health. 2021;9:623904. Published 2021 Feb 3. doi:10.3389/fpubh.2021.623904

  1. The study mainly covers European nations and Israel. What might be its implications for poorer nations? 

Added in the INTRODUCTION

Low- and middle-income countries have faced many challenges in controlling COVID-19 pandemic; in these countries healthcare resources are limited,  SARS COV 2  testing  is conducted on a limited scale and treatment options are few. Very often there is no vaccine. Only Low-cost solutions remain for the prevention, diagnosis, and treatment of SARS-CoV-2. 

Aziz, Asma B., et al. "Integrated control of COVID-19 in resource-poor countries." International Journal of Infectious Diseases 101 (2020): 98-101.

  1. The percentage of rural location of investigated samples is so different across countries, from Romania to Greece. But what this difference may mean to the investigated health services?

            As stated in the limitations: The limitations include the relatively low response rate (22.0%) with considerable differences between countries. Moreover, the survey was based on a self-selecting sample, which comes with inherent bias. Also, the sample was obtained differently depending on the participating country. Moreover, the data collection was carried out in several months therefore covering different phases of the pandemic. In addition, rural practices may be differently characterized in different countries, which may influence the results of our study.  Despite these limitations which don't allow comparison between countries,  the high number of responses is a strength.

  1. The authors mentioned intimate partner violence (IPV) and domestic violence, but didn’t provide any detailed discussion on the violence and the COVID-19 pandemic situation. Other psychological and mental issues weren’t widely covered either. Should the study cover/report more such information?

Our study was not aimed at evaluating the presence of mental problems in our patients but to find out to what extent GPS used to check with patients  the potential consequences of psychological and mental issues, and if there was a different attitude of the GPs before and after the pandemic.

The question in the survey  was To what extent have you checked with patients to determine if they (in)directly experienced family violence since the COVID-19 pandemic?

  1. For analysis, would authors consider adopting more methods to reflect the possible interaction between the reported factors? As stated in the limitations paragraph we had a relatively low response rate (22.0%) with considerable differences between countries. Moreover, the survey was based on a self-selecting sample, which comes with inherent bias. Also, the sample was obtained differently depending on the participating country. Moreover, the data collection was carried out in several months therefore covering different phases of the pandemic. In addition, rural practices may be differently characterized in different countries, which may influence the results of our study. This does not allow sophisticated speculations among the reported factors but mainly descriptive statistics leaving to future studies to explore more in depth the interactions between these factors.

  1. The study reported “Higher levels of rural broadband exist in Europe, and in some countries, rural broadband coverage exceeds urban, indicating that 322 other factors are at play here, possibly health literacy and older populations” (P11, 321-323). Why didn’t the study provide more such information? Even though they were not covered by the survey, the authors can manage to get them from second-hand sources.

This part has been rephrased

A part the digital divide, other factors are at play here, possibly health literacy and older populations

References

Nair, S. C., Satish, K. P., Sreedharan, J., Muttappallymyalil, J., & Ibrahim, H. (2020). Improving Health Literacy Critical to Optimize Global Telemedicine During COVID-19. Telemedicine and e-Health, 26(11), 1325-1325.

Finally, I don’t have public health practice experience and my academic training is health communication. So, if some of my suggestions, which are regular in a health communication setting, may not be acceptable to public health scholarship, I am fine that the authors insist on their own preferences after providing appropriate explanations in responses or in manuscript revision. 

Round 2

Reviewer 1 Report

Dear authors: complete details regarding the strategies to follow based on the strengths and weaknesses found in the comparative studies based  on the global vision of the countries analyzed

Author Response

Thank you very much for your suggestion

Reviewer 1

Yes

Can be improved

Must be improved

Not applicable

Does the introduction provide sufficient background and include all relevant references?

(x)

( )

( )

( )

Are all the cited references relevant to the research?

(x)

( )

( )

( )

Is the research design appropriate?

(x)

( )

( )

( )

Are the methods adequately described?

(x)

( )

( )

( )

Are the results clearly presented?

(x)

( )

( )

( )

Are the conclusions supported by the results?

( )

(x)

( )

( )

Comments and Suggestions for Authors

Dear authors: complete details regarding the strategies to follow based on the strengths and weaknesses found in the comparative studies based  on the global vision of the countries analyzed.

We have added in the conclusion:

Particular emphasis should be placed on reducing inequalities in access to modern forms of remote communication with the use of computer and ICT techniques. Improvement in this area is needed in both urban and rural practices but in the latter, however, there is a greater need. Furthermore, it appears from our data that GP practices, regardless of urban/rural location, responded according to recommendations during the COVID-29 pandemic, but differences in patient populations may require differential responses which need to be factored in. Our findings show the existence of certain issues that could impact patient safety in rural areas more than in urban areas due to the underlying differences in population profile and supports. These could be used to plan the organization of care for similar future pandemic situations.

Reviewer 2 Report

I have viewed the revised version and I believe after revision, the current version has met the criteria of publication in IJERPH

Thank you

Author Response

Thank you for having accepted the revised release. A mother tongue coauthor has performed the editing of the English spelling and style.

Reviewer 2

Open Review

English language and style

( ) English very difficult to understand/incomprehensible
( ) Extensive editing of English language and style required
( ) Moderate English changes required
(x) English language and style are fine/minor spell check required
( ) I don't feel qualified to judge about the English language and style

I have viewed the revised version and I believe after revision, the current version has met the criteria of publication in IJERPH

Thank you for having accepted the revised release. A mother tongue coauthor has performed the editing of the English spelling and style.